# In Vitro Antimicrobial Effect of Novel Electrospun Polylactic Acid/Hydroxyapatite Nanofibres Loaded with Doxycycline

**DOI:** 10.3390/ma15186225

**Published:** 2022-09-07

**Authors:** Vlad Andrei, Nicodim Iosif Fiț, Ioana Matei, Réka Barabás, Liliana Antonela Bizo, Oana Cadar, Bianca Adina Boșca, Noémi-Izabella Farkas, Laura Marincaș, Dana-Maria Muntean, Elena Dinte, Aranka Ilea

**Affiliations:** 1Department of Oral Rehabilitation, Faculty of Dentistry, Iuliu Hațieganu University of Medicine and Pharmacy, 400012 Cluj-Napoca, Romania; 2Department of Microbiology, Mycology and Immunology, Faculty of Veterinary Medicine, University of Agricultural Science and Veterinary Medicine, 400372 Cluj-Napoca, Romania; 3Department of Chemistry and Chemical Engineering of Hungarian Line of Study, Faculty of Chemistry and Chemical Engineering, 400028 Cluj-Napoca, Romania; 4Department of Chemical Engineering, Faculty of Chemistry and Chemical Engineering, Babeș-Bolyai University, 400028 Cluj-Napoca, Romania; 5INCDO-INOE 2000, Research Institute for Analytical Instrumentation, 400293 Cluj-Napoca, Romania; 6Department of Morphological Sciences, Faculty of Medicine, “Iuliu Hațieganu” University of Medicine and Pharmacy, 400012 Cluj-Napoca, Romania; 7Department of Chemistry, Faculty of Chemistry and Chemical Engineering, Babeș-Bolyai University, 400028 Cluj-Napoca, Romania; 8Department of Pharmaceutical Technology and Biopharmaceutics, Faculty of Pharmacy, “Iuliu Hațieganu” University of Medicine and Pharmacy, 400012 Cluj-Napoca, Romania

**Keywords:** polylactic acid, nano-hydroxyapatite, doxycycline, electrospinning, nanofibres, antimicrobial, periodontitis

## Abstract

The present study aimed to assess the in vitro antimicrobial effects of a novel biomaterial containing polylactic acid (PLA), nano-hydroxyapatite (nano-HAP) and Doxycycline (Doxy) obtained by electrospinning and designed for the non-surgical periodontal treatment. The antimicrobial activity of two samples (test sample, PLA-HAP-Doxy7: 5% PLA, nano-HAP, 7% Doxy and control sample, PLA-HAP: 5% PLA, nano-HAP) against two periodontal pathogens—*Aggregatibacter actinomycetemcomitans* and *Porphyromonas gingivalis*—was assessed using the Kirby–Bauer Disk Diffusion Susceptibility Test and compared with the effect of four antibiotics used as adjuvants in periodontal therapy: Amoxicillin, Ampicillin, Doxy and Metronidazole. The test sample (embedded with Doxy) showed higher inhibitory effects than commonly used antibiotics used in the treatment of periodontitis, while the control sample showed no inhibitory effects. Moreover, significant differences were observed between the inhibition zones of the two samples (*p* < 0.05). The Doxy-loaded PLA nanofibres had an antimicrobial effect against the periodontal pathogens. Based on these results, the novel biomaterial could be a promising candidate as adjuvant for the non-surgical local treatment in periodontitis.

## 1. Introduction

Periodontitis is a chronic, inflammatory, multifactorial disease, initiated by the dysbiotic microbial biofilm, characterised by the progressive destruction of the tooth-supporting periodontal apparatus [1,2].

It is estimated that the oral microbiome in adult patients consists of approximately 700 different bacterial species, half of these being concomitantly present concomitantly at any time. The shifting of the ratio between pathogenic and beneficial bacterial species, caused by the accumulation of the biofilm, initiates periodontal inflammation, leading to destruction of the ligament apparatus and resorption of the alveolar bone. However, one of the key factors in the initiation and the progression of periodontitis is the high concentration of pathogenic bacteria [3]. Two periodontal pathogenic bacteria, *Aggregatibacter actinomycetemcomitans* and *Porphyromonas gingivalis,* have been linked to unfavourable disease progression and poor therapeutic results.

*A. actinomycetemcomitans* is a highly pathogenic bacterium involved in periodontal disease [4,5]. The increased pathogenic potential is due to production of endotoxins that induce rapid degradation of periodontal tissues in young individuals [6]. These endotoxins alter the activity of various types of leukocytes, leading to the release of pro-inflammatory mediators and interleukin-1β, which activates osteoclasts and results in bone tissue resorption [7,8,9].

*P. gingivalis* is also involved in pathogenesis of periodontal disease [10,11]. *P. gingivalis* has been shown to have an arsenal of virulent factors such as lipopolysaccharides, proteases, and fimbriae. These factors trigger multiple mechanisms: alter the innate immunity leading to oral dysbiosis, induce cellular senescence both directly, by invasion of keratinocytes, endothelial cells, and dendritic cells, and indirectly by stimulating the secretion of inflammatory exosomes and thus alter the functions of cells involved in local immune defence. Moreover, the released proteases can paralyse chemokines by IL-8 degradation, thus altering the activity of the complement system and promoting dysbiosis by persistence of pathogenic bacteria in the oral cavity [12,13].

Both *A. actinomycetemcomitans* and *P. gingivalis* have been associated with aggressive periodontitis in young patients, causing severe destructions of the periodontal tissues, over a short period of time after the disease onset [3,14].

Since efficient periodontal therapy involves synergistic control of local and general risk factors, the crucial step in the non-surgical treatment is the removal of subgingival plaque and calculus, in order to reduce the local inflammation and to improve clinical markers, such as probing depth and gingival bleeding [2].

Given the infectious nature of periodontitis, the use of antibiotics for local application and general administration has been proposed and studied [2,14]. However, due to the concerns regarding the patient’s health and the impact on public health, routine use of systemic antibiotics is not recommended [2,15]. Locally delivered antibiotics have the potential to eliminate the general side effects [16], but their use is currently limited due to the difficult application and rapid clearance from the periodontal pockets [2,16].

Furthermore, due to excessive use of antibiotics in skin and respiratory infections, many pathogenic bacteria have developed antibiotic resistance (AR). Untreatable infections caused by multidrug-resistant strains of bacteria are responsible for high treatment costs and increased mortality rates [17,18,19]. Understanding the emergence of AR is the key to improving antibiotic treatment and finding new alternatives and formulas to stop this phenomenon [19,20].

The use of polymers in medicine has been researched, with polylactic acid (PLA) emerging as a preferred candidate in medical use, due to its biocompatibility and mechanical characteristics. PLA has proven a slow resorption when placed in situ, which occurs through hydrolytic degradation, and does not produce any compounds, thus making it an ideal candidate for local drug administration [21]. Its use as a drug-delivering system has been investigated, due to the polymer’s drug loading capabilities and ease of manufacture [22]. Doxycycline (Doxy) is a semi-synthetic derivate of Tetracycline, used in the adjunctive treatment of periodontitis [23]. The antibiotic has been used for both systemic administration and local applications, having shown clinical improvements [24,25]. Nano-hydroxyapatite (nano-HAP) is a bioceramic material with a similar structure to bone andhard dental tissues. Moreover, nano-HAP has shown good biocompatibility and affinity to polymers, thus being used for tissue regeneration and tissue engineering [26].

The aim of the present study is to assess the in vitro antimicrobial effect of a previously characterised and evaluated novel biomaterial based on PLA nanofibres and nano-HAP, loaded with Doxy obtained by electrospinning against two periodontal pathogens: *A. actinomycetemcomitans* and *P. gingivalis*.

## 2. Materials and Methods

### 2.1. Sample Characterisation

Two different samples of nanofibres were obtained using the electrospinning technique. The control sample was based on a 5% solution of PLA and nano-HAP. The test sample was based on the same PLA and nano-HAP solution, loaded with 7% Doxy by physical adsorption. The PLA-HAP and PLA-HAP-Doxy7 (Doxy-loaded HAP sample encapsulated in PLA) nanofibres containing 5% PLA, a PLA/HAP ratio of 0.8 and 7 wt.% Doxy were prepared by electrospinning using a Fluidnatek LE-50 benchtop line (Bioinicia S.L., Valencia, Spain) by applying a voltage, flow-rate and tip-to-collector distance set at 27 kV, 1 mL/h and 15 cm as described in details in our previous paper [27]. The PLA-based nanofibres were characterised by Fourier transform infrared (FT-IR) spectroscopy, thermogravimetry-differential thermal analysis (TG-DTA) and scanning electron microscopy (SEM). Moreover, the in vitro drug release studies in phosphate buffer (PBS) and simulated body fluid (SBF) recommended PLA-HAP-Doxy7 sample as potential drug delivery system [27]. The manufacturing and characterisation process of the nanofibres are represented schematically in Figure 1.

Studies regarding the amount of Doxy in the tested nanofibres showed a content of 8.40 ± 0.10 µg Doxy/1 mg nanofibres, with the antibiotic uniformly distributed in the mass of nanofibres. Doxy concentrations were calculated using a calibration curve that proved to be linear between 5 and 100 µg/mL, by a validated HPLC-UV method, at a wavelength of 365 nm.

The antimicrobial activity was evaluated in anaerobiosis, under strictly controlled temperature of 37 ± 0.1 °C [5,28]. The bacterial strains *A. actinomycetemcomitans* (ATCC 29522) and *P. gingivalis* (ATCC 33277) were lyophilised and were regenerated on anaerobic 5% sheep blood agar mediums (NutriSelect^®^ Plus, Merck KGaA, Darmstadt, Germany), and the first subcultures of the cell strains were used for further experiments.

### 2.2. Control Sensibility Tests

The control sensibility tests on the pathogens used four antibiotics commonly administered as adjuvants in periodontal treatment: Amoxicillin 10 µg, Ampicillin 25 µg, Doxycycline 10 µg and Metronidazole 5 µg (Bioanalyse, Ankara, Turkey), in the form of standardised micro compresses [3].

The antimicrobial activity of the samples was tested using the diffusimetric method, based on the technique described in the Kirby–Bauer Disk Diffusion Susceptibility Test Protocol. The inoculum was obtained from the seventh day microbial cultures, with a density of 0.5 standard McFarland (10^6^ UFC/mL). This inoculum was dispersed on the testing medium using a sterile cotton swab [29,30].

### 2.3. Sample Preparation

The samples were prepared for testing by adding 0.0100 g of each compound in 500 μL sterile saline solution (NaCl 0.9%). Next, the solutions were individually homogenised at 250 rpm for 60 min, at a constant temperature of 22 ± 2 °C. Subsequently, the samples were stored in sterile Eppendorf tubes.

### 2.4. Microbiological Assessment

Sterile micro compresses of 10 mm diameter were then soaked with 30 μL of each obtained compound. Therefore, the amount of Doxy contained in the tested nanofibre mass was 5.04 ± 0.01 µg, reconstituted in the form of 30 μL of sterile saline solution. Fifteen minutes after the Petri dishes were flooded with the inoculum, three different micro compresses were radially placed on the medium. The dishes were anaerobically incubated using a GENbag-GENbox type system (bioMérieux S.A., Marcy l’Etoile, France), at a controlled temperature of 37 °C. The anaerobiotic state was obtained using an anaerobiosis atmosphere generation system (Anaerocult™, Merck KGaA, Darmstadt, Germany, and further validated using dedicated colour-changing strips (Anaerotest™, Merck KGaA, Darmstadt, Germany)).

The results were recorded after seven days. The diameters of the resulting inhibition zones for both the controls and test samples were measured using an electronic calliper gauge, and the values were expressed in millimetres with two decimals. All tests were run in triplicate, and an average value of the inhibition zones was calculated. Data were statistically analysed, and the graphs were generated using the GraphPad Prism Software (GraphPad Software, Inc., San Diego, CA, USA).

## 3. Results

After seven days, the obtained colonies were optimal, having the disposition and morphology characteristic for the tested species (Figure 2).

### 3.1. Sample Characterisation

PLA-HAP and PLA-HAP-Doxy7 samples were previously characterised by FT-IR, TGA and SEM. The in vitro drug release studies recommended PLA-HAP-Doxy7 sample as a potential drug delivery system, since the drug release profile in SBF displayed the highest concentrations of released Doxy over a long period of time (100 h) [27]. FT-IR spectroscopy evidenced the characteristic bands of PLA, namely, stretching frequencies of C=O, –CH_3_ asymmetric and C–O at 1749, 2998 and 1079 cm^−1^, and bending frequencies of –CH_3_ asymmetric and symmetric at 1452 and 1361 cm^−1^, respectively. The presence of Doxy was confirmed by the characteristic bands in the range of 3500–3000 cm^−1^ (νOH and νNH), 1610 cm^−1^ (amide band I) and 1570 cm^−1^ (amide band II), and C=O and C=C stretches in the range of 1700–1600 cm^−1^. According to SEM, the average diameter of PLA-HAP nanofibre (310 ± 12 nm) increased significantly to 363 ± 24 nm upon the addition (encapsulation) of Doxy in case of PLA-HAP-Doxy7 [27]. The average diameter was assessed using the ImageJ software and expressed as mean ± standard deviation. Hence, a significant increase in the average fibre diameters was observed for the PLA-HAP-Doxy sample compared to the PLA-HAP sample.

The drug release studies on the PLA-HAP and PLA-HAP-Doxy7 samples showed a sustained release in both investigated media, i.e., SBF and PBS as the release was constrained by the encapsulation of the drug-loaded HAP in the polymer fibres. The release of Doxy from the PLA-HAP-Doxy7 sample displayed a slowly, but increasing dissolution tendency in SBF, reaching 6.08% after 96 h. However, the Doxy release kinetics were not explained by any of the studied mathematical models (zero-order kinetics, first-order kinetics, Higuchi, Hixon–Crowell and Korsmeyer–Peppas model), regardless of the investigated medium. A possible explanation could be that the drug release from nanofibres is influenced by many factors such as nature of drug, polymer, additives, solvent, diffusion of the drug through the polymer matrix, etc. [27]. Consequently, our previous results regarding the investigated electrospun polylactic acid/hydroxyapatite nanofibre loaded with Doxycycline (PLA-HAP-Doxy7) required further in vitro studies in order to confirm their potential application as a drug delivery system.

### 3.2. Control Sensibility Tests

The control antimicrobial tests on the bacterial strains showed an increased sensibility to the four antibiotics, demonstrated by the presence of the inhibition zones. The largest inhibition zone was induced on *A. actinomycetemcomitans* by Metronidazole (26.65 mm, SD ± 0.5160), and on *P. gingivalis* by Doxycycline (28.74 mm, SD ± 1.242) (Table 1).

### 3.3. Microbiological Assessment

The sensibility tests obtained from the primary subcultures, using the same incubation environments, allowed for the evaluation of the antimicrobial effects of the proposed materials and control antibiotics on the two bacterial strains (Table 2).

The test sample (PLA-HAP-Doxy7) produced an average inhibition zone of 32.95 mm (SD ± 0.7199) on *A. actinomycetemcomitans,* and the control sample (PLA-HAP) produced an inhibition zone of 11.87 mm (SD ± 0.4120). The Student *t* test revealed statistical differences between the two samples (Table 2).

The test sample (PLA-HAP-Doxy7) induced an average inhibition zone of 33.10 mm (SD ± 1.537) on *P. gingivalis,* whereas the control sample (PLA-HAP) induced an inhibition zone of 11.83 mm (SD ± 0.5393). The Student *t* test revealed significant differences between the two samples (Table 2).

Significant differences were also observed between the inhibition zones induced by the novel biomaterial embedded with Doxy and the standardised micro compresses with Doxy on both *A. actinomycetemcomitans* (*p* < 0.0001) and *P. gingivalis* (*p* = 0.0189), even though the concentration of the sample test was half that in the standardised control.

## 4. Discussion

The oral cavity is considered, from an ecological microbiological point of view, an “open growth system”, due to its constant flux of ingestion and excretion of microorganisms [3]. The microorganisms found in the oral cavity are organised in a complex structure of salivary glycoproteins and extracellular microbial products, which not only promotes the bacterial survival, but also facilitates their adherence to the teeth, prosthetics and gums [31]. Due to proliferation of pathogenic bacteria, the accumulation of plaque extends subgingivally and, eventually, can even infiltrate the soft periodontal pocket wall, where the mechanical treatment has little to no effect [3].

The non-surgical periodontal treatment consists in scaling the periodontal pockets and planning the root surfaces of affected teeth, with the end goal of removing the biofilm, reducing the bacterial load and controlling the periodontal inflammation [32]. However, certain anatomic particularities, such as the presence of very deep periodontal pockets, irregularities of the root surface or presence of negative anatomic morphologies (e.g., furcation areas) may hinder the success of the non-surgical therapy and could increase the risk for reinfection [16,32].

Given these challenges in the periodontal therapy, the administration of antibiotics was proposed for more efficient elimination of periodontal pathogens and prevention of the reinfection of dental pockets [31]. Two main routes are recommended for the adjuvant antibiotic therapy in periodontitis: the systemic administration and local application [32].

The main advantage of the systemic administration of antibiotics is their ability of reaching not only the periodontal pockets, but also the other areas of the oral cavity that may constitute reservoirs for reinfections, such as the dorsal surface of the tongue, the epithelium on the floor of the mouth and the tonsils [3,15,32]. On the other hand, since the antibiotic’s distribution is wider in other areas of the body, only a small concentration is obtained in the periodontal tissues [15,32]. Furthermore, the systemic administration of the antibiotics is dependent on patient compliance and may cause more frequent general adverse reactions, compared with local applications [15,32].

The locally applied antibiotics enable reaching higher concentrations of active substances in the periodontal tissues, while avoiding the general complications that systemic administration may present [15,16]. However, the prolonged delivery of the therapeutic substances has proven to be difficult, as the half-life of antibiotics placed inside periodontal pockets was estimated to be approximately one minute. Moreover, the crevicular fluid constantly secreted into the sulcus and periodontal pockets is responsible for the removal of any liquids or gels that are applied locally [32]. Therefore, the development of a product capable of slowly releasing the antibiotic locally could increase the efficiency of periodontal treatment, with better outcomes in severe or aggressive cases.

The two samples used in the present study were based on nanofibres obtained through electrospinning, a technique that uses high-voltage electric current and pressure to transform a polymeric solution into nano-fibres [33]. PLA was chosen due to the biocompatibility, slow resorption in situ and lack of toxic by-products after resorption [21]. Addition of nano-HAP to the polymeric material was considered due to its osteoconductive properties, required for the regeneration of lost periodontal tissues [26]. The biomaterial was embedded with Doxy, an antibiotic in the tetracycline class with proven efficiency against both aerobic and anaerobic bacteria [23].

Regarding the present study, one of the main challenges was the cultivation of the two bacterial species involved in the onset and progression of periodontitis. Both *A. actinomycetemcomitans* and *P. gingivalis* were used as pure culture strains and required special incubation methods, regarding the anaerobiosis and controlled temperature. From a practical point of view, this translated in more difficult manipulation during the experiment [28].

The control antibiotics tested on the two pathogens have proven the efficiency of Doxy, in comparison with other commonly used antibiotics. The antimicrobial effect of Doxy on *A. actinomycetemcomitans* was lower compared with Metronidazole (inhibition areas of 17.83 ± 0.1528 mm and 26.65 ± 0.5160 mm, respectively), but higher on *P. gingivalis* than any of the other antibiotics (inhibition area of 28.74 ± 1.242 mm). The clinical impact of these findings resides in the high frequency of recurrence of *P. gingivalis* in aggressive forms of periodontitis (79.6% of cases), with high bone resorption rates in young individuals and recurrent reinfections after treatment [3]. In these cases, the local administration of Doxycycline may prove beneficial and could prevent relapses after treatment.

One aspect that must be taken into consideration is the behaviour of the two pathogens in their pure, solitary state, compared to the behaviour in the biofilm. The oral biofilm not only plays a role in the nutrition and adhesion of the bacteria on the oral surfaces, but also may protect bacteria from the penetration of antibiotics, thus reducing the therapeutic effect [3]. This aspect underlines the need for a synergic mechanical and chemical treatment, and the need for an in vivo evaluation of the developed novel biomaterial, for thorough evaluation of its effect on the biofilm.

The results obtained in the present study show that, out of the two samples of novel biomaterial tested on the two bacterial strains, the sample containing Doxycycline had a good antibacterial effect. The sample containing only PLA and nano-HAP had no inhibitory effect on the two pathogens. PLA is used as a carrier for Doxy and nano-HAP, allowing for the slow release of the substances thorough its resorption. Thus, the lack of antibacterial properties of PLA is expected.

The increased antibacterial effect of the tested nanofibres, loaded with half the Doxy concentration (5.04 µg) of the Doxy micro compresses (10 µg), was demonstrated by the larger diameter of the inhibition zones. Thus, the diameter of the inhibition zone in the case of nanofibres loaded with Doxy was 32.95 ± 0.7199 mm, compared with 17.83 ± 0.1419 mm for Doxy control, for A. actinomycetemcomitans, and 33.10 ± 1.537 mm in the case of nanofibres loaded with Doxy, compared with 28.74 ± 1.242 mm obtained for control, on *P. gingivalis*.

Studies conducted by Bhat et al., revealed, by the analysis of 40 isolates of *A. actinomycetemcomitans* from patients, a sensitivity to Doxycycline, with a minimal inhibitory concentration (MIC) of 0.064–8 μg/mL [34]. Another study conducted in Yemen on 30 strains of *A. actinomycetemcomitans* isolated from patients with localised aggressive periodontitis showed a low susceptibility of strains to Doxycycline, 46.7% being considered sensitive and 53.3% moderately sensitive, while observing a sensitivity of 100% for Cefotaxime and Ceftriaxone [35].

These studies showed that *A. actinomycetemcomitans* strains may have different sensitivities depending on the patients from whom the bacteria had been obtained and the population studied. It is known that Fluoroquinolones and Cephalosporins have a good effect on the bacterial flora in the oral cavity, respiratory tract, and digestive tract [35,36]. Similar results have been reported in many countries where there has been an increase in the resistance of *A. actinomycetemcomitans* strains, although several studies showed the excellent effect of Doxycycline against bacterial biofilm [35,37,38].

The novel biomaterial has been previously characterised, and the results were published. The release mechanisms of the material have been investigated, including the desorption analyses conducted in two different solutions: simulated body fluid and phosphate buffer solution [27]. These conditions could not be simulated on the Petri dishes used in the present studies. Thus, it is expected that the in vitro desorption curves could differ from the ones obtained in vivo.

Three different concentrations of Doxy (3%, 7% and 12%) were used for obtaining the materials that were previously characterised. According to the results, prolonged release of the antibiotic was achieved for the samples containing 3% and 7% Doxy [27]. The sustained release profile of PLA-HAP-Doxy7 sample in SBF, due to the encapsulation of the drug into the PLA-HAP nanofibres, is considered to be of greatest interest [27]. Consequently, for the present study, we chose this sample due to the constant, extended release rate of Doxy instead of the immediate release of the drug. The sustained release could allow for the delivery of Doxy at a designed rate for a prolonged period of time. Thus, the 7% Doxy-loaded sample was chosen for the current in vitro evaluation.

A limitation of the present study resides in the impossibility to evaluate the effects that nano-HAP has on the healing of affected periodontal tissues. HAP has long proven its biocompatibility and role in modulating the host response in osteogenesis [26]. Moreover, since PLA degradation occurs over time, by the activity of hydrolytic enzymes, the full potential of the novel biomaterial as an adjuvant for the non-surgical treatment of periodontitis should be investigated in vivo. The developed material could be of use following the manual and ultrasonic debridement of periodontal pockets, in cases of aggressive bacterial populations, recurring infections or previous unsatisfactory therapeutic results. The material could be placed in the treated pockets, releasing the Doxy over a longer period, and helping with the stabilisation of the blood cloth, conditions needed for the reduction of the pocket depth and improvement of the clinical parameters.

## 5. Conclusions

Both *A. actinomycetemcomitans* and *P. gingivalis* were successfully cultivated in vitro, using special anaerobic environments and controlled temperatures. The two bacterial strains were susceptible to the antibacterial effects of Amoxicillin, Ampicillin, Doxycycline, and Metronidazole. The evaluation of the test sample consisting of nanofibres developed through the electrospinning technique (based on PLA nanofibres, nano-HAP, Doxy) on the two bacterial strains proved a good antimicrobial effect. The inhibitory effect of biomaterial samples containing Doxycycline was comparable to that of other antibiotics commonly used for periodontal treatment, and higher than micro compresses control. The samples without Doxy, containing only PLA nanofibres and nano-HAP, showed no antimicrobial effects on the two pathogens.

## 6. Patents

Patent pending with registration number A/00385/05.07.2022 at Romanian State Office for Inventions and Trademarks.

## Figures and Tables

**Figure 1 materials-15-06225-f001:**
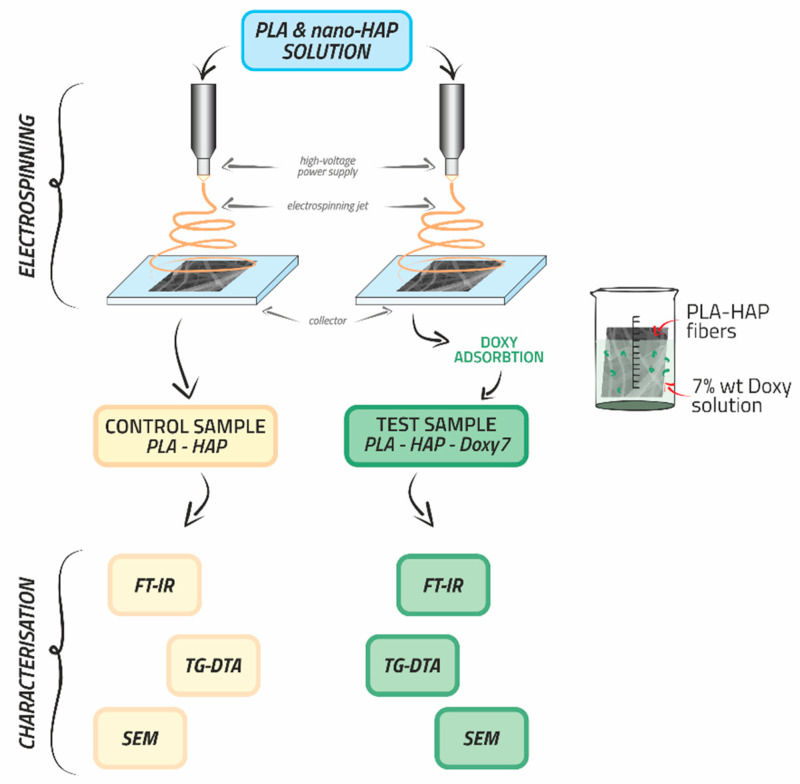
Schematic representation of the technological manufacturing process of the nanofibres through electrospinning and their characterisation.

**Figure 2 materials-15-06225-f002:**
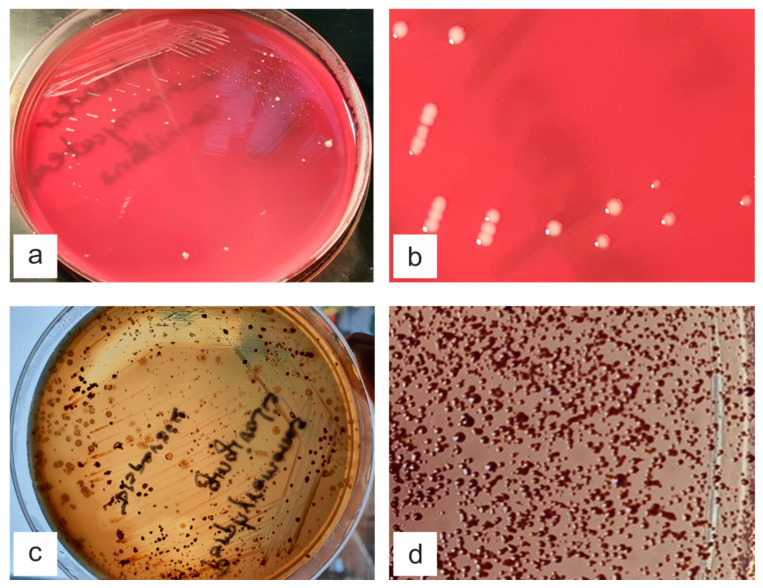
The colonies of strains *A. actinomycetemcomitans* (ATCC 29522) and *P. gingivalis* (ATCC 33277) after seven days of incubation at controlled temperature of 37 ± 0.1 °C in anaerobiosis, in 5% sheep blood agar medium: (**a**) *A. actinomycetemcomitans* culture on blood agar medium: Petri dish overview; (**b**) *A. actinomycetemcomitans* colonies on blood agar medium: detailed image of the colonies; (**c**) *P. gingivalis* culture on blood agar medium: Petri dish overview; (**d**) *P. gingivalis* colonies on blood agar medium: detailed image of the colonies exhibiting a brown colour and complete haemolysis.

**Table 1 materials-15-06225-t001:** Mean values of inhibition zone diameters (expressed in mm ± SD—standard deviation) induced by the control antibiotics on *A. actinomycetemcomitans* and *P. gingivalis*.

Bacterial Species	Control Antibiotics			
	Amoxicillin	Ampicillin	Doxycycline	Metronidazole
*A. actinomycetemcomitans*	13.37 ± 0.1484	17.11 ± 0.1528	17.83 ± 0.1419	26.65 ± 0.5160
*P. gingivalis*	25.90 ± 1.132	21.24 ± 0.7454	28.74 ± 1.242	27.75 ± 0.7094

**Table 2 materials-15-06225-t002:** Mean values of inhibition zone diameters (expressed in mm ± SD—standard deviation) for the tested biomaterial samples on *A. actinomycetemcomitans* and *P. gingivalis*.

Bacterial Species	Samples		
	Test (PLA-HAP-Doxy7)	Control (PLA-HAP)	*p* Value
*A. actinomycetemcomitans*	32.95 ± 0.7199	11.87 ± 0.4120	*p* < 0.0001
*P. gingivalis*	33.10 ± 1.537	11.83 ± 0.5393	*p* < 0.0001

## Data Availability

Not applicable.

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
