# Peer review of "In Vitro Antimicrobial Effect of Novel Electrospun Polylactic Acid/Hydroxyapatite Nanofibres Loaded with Doxycycline"

_materials, 2022, doi:10.3390/ma15186225_

Round 1
Reviewer 1 Report
1. In page 2-line no92-93: What mechanical properties of PLA attracted in biomedical applications?
2. In methodology section: Electrospinning is used but what setting parameters have been used is not explained. I think this is necessary.
3. For well understanding and better visibility materials and method should be explained in a different sub section. Similarly, testing and characterization should have a subheading.
4. In result and discussion section: I am not clear how this paper is designed.
Chemical characterisation for different materials is not explained. For instance, FTIR??

Author Response
Dear sir/ madam,
We would like to firstly thank you for agreeing to review our submitted paper to the special issue in the journal Materials, “Advanced Smart Biomaterials and Techniques for Oral, Hard Tissue Engineering and Regeneration”.
We have analyzed your comments and suggestions to the manuscript and made modifications accordingly. Thus, the following alterations were made:
- In page 2-line no92-93: What mechanical properties of PLA attracted in biomedical applications?
The following sentence was added, to explain the thought process for choosing PLA as a carrier (lines 94-96):
“PLA has proven a slow resorption when placed in situ, which occurs through hydrolytic degradation, and does not produce any compounds, thus making it an ideal candidate for local drug administration [21].”
- In methodology section: Electrospinning is used but what setting parameters have been used is not explained. I think this is necessary.
The used parameters are stated in the methodology section, lines 116-118:
“Doxy were prepared by electrospinning using a Fluidnatek LE-50 benchtop line (Bioinicia S.L., Valencia, Spain) by applying a voltage, flow-rate and tip-to-collector distance set at 27 kV, 1 mL/h and 15 cm”
- For well understanding and better visibility materials and method should be explained in a different sub section. Similarly, testing and characterization should have a subheading.
For an easier understanding of the text article, the following subheadings were added in the materials and methods section:
2.1. Sample characterisation (line 110)
2.2. Control sensibility tests (line 134)
2.3. Sample preparation (line 145)
2.4. Microbiological assessment (line 150)
Moreover, the following three subheadings were added in the results section of the paper:
3.1. Sample characterisation (line 178)
3.2. Control sensibility tests (line 207)
3.3. Microbiological assessment (line 214)
- In result and discussion section: I am not clear how this paper is designed.
The discussion chapter of the paper begins by explaining the complex nature of the oral cavity from a microbiological point of view, highlighting the difficulties that periodontists and dental professionals need to tackle in treating periodontitis. Then, it continues to highlight current trends in antibiotic administration in periodontitis patients, underlining the need for a material capable of successfully delivering antibiotics locally. After detailing the difficulties in working with anaerobic bacteria, the focus shifts to the sensitivity tests and the reasons why Doxycycline was chosen. Furthermore, the results in the present study are explained, detailed, and compared with relevant information in the literature. Lastly, the limitations of the present study are emphasized and the need for further, in vitro evaluation is stated.
Chemical characterisation for different materials is not explained. For instance, FTIR??
The following sentences were added:
Lines 183 – 188: “FT-IR spectroscopy evidenced the characteristic bands of PLA, namely stretching frequen-cies of C=O, –CH3 asymmetric and C–O at 1749, 2998 and 1079 cm−1, and bending fre-quencies of –CH3 asymmetric and symmetric at 1452 and 1361 cm−1, respectively. The presence of Doxy was confirmed by the characteristic bands in the range of 3500–3000 cm−1 (νOH and νNH), 1610 cm−1 (amide band I) and 1570 cm−1 (amide band II), and C=O and C=C stretches in the range of 1700-1600 cm−1.”
Lines 190 – 193: “The average dimeter was assessed using the ImageJ software and expressed as mean ± standard deviation. Hence, a significant increase of the average fiber diameters was ob-served for PLA-HAP-Doxy sample compared to PLA-HAP sample.”
We are looking forward to the assessment of the modified manuscript.
Sincerely,
The authors

Reviewer 2 Report
The manuscript by Andrei et al. evaluates the in vitro antimicrobial effect of novel electrospun polylactic 2 acid/ hydroxyapatite nanofibers, that were loaded with Doxycycline. It fits well to the thematic issue of then Materials SI it is submitted too (Advanced Smart Biomaterials and Techniques for Oral, Hard Tissue Engineering and Regeneration). The following points have to be addressed, as mentioned below:
1. The manuscript is a continuation of the work of some of the authors from 2022 (ref 27), where the design of the PLA/nanoHA composite with Doxy was evaluated and determined. As such, the materials characterization data from the samples investigated in vitro in this article is mostly missing, and authors are referred to the data and discussion in previous article. Nevertheless, in the present work authors evalauate only 2 morphologies of samples: the Doxy sample and the reference. Some minimal characterization data can be still shown (remeasured, or adapted and cited from author's previous work), to give the readers a more clear overview. This also as the discussion from ln 169-171 (and in results and discussion ln 312-330) is quite comprehensive and detailed, but the addition of basic characterization for the investigated samples can add more depth.
2. Abstract can be improved, removal of Background, Method, etc., to improve readability and make it more attractive for readers to follow.
3. Other minor things: a) when referring to samples in Table, should be clear which is the tested and which is the control sample, b) minor English corrections are needed. c) Introduction: the emphasis of the aim of the present study can be improved - design of biomaterial was already optimized in a previous work, and now the focus in for the optimal sample, the in vitro antimicrobial effect.
Author Response
Dear sir/ madam,
We would like to firstly thank you for agreeing to review our submitted paper to the special issue in the journal Materials, “Advanced Smart Biomaterials and Techniques for Oral, Hard Tissue Engineering and Regeneration”.
We have analyzed your comments and suggestions to the manuscript and made modifications accordingly. Thus, the following alterations were made:
- The manuscript is a continuation of the work of some of the authors from 2022 (ref 27), where the design of the PLA/nanoHA composite with Doxy was evaluated and determined. As such, the materials characterization data from the samples investigated in vitro in this article is mostly missing, and authors are referred to the data and discussion in previous article. Nevertheless, in the present work authors evalauate only 2 morphologies of samples: the Doxy sample and the reference. Some minimal characterization data can be still shown (remeasured, or adapted and cited from author's previous work), to give the readers a more clear overview. This also as the discussion from ln 169-171 (and in results and discussion ln 312-330) is quite comprehensive and detailed, but the addition of basic characterization for the investigated samples can add more depth.
The following sentences were added:
Lines 183 – 188: “FT-IR spectroscopy evidenced the characteristic bands of PLA, namely stretching frequen-cies of C=O, –CH3 asymmetric and C–O at 1749, 2998 and 1079 cm−1, and bending fre-quencies of –CH3 asymmetric and symmetric at 1452 and 1361 cm−1, respectively. The presence of Doxy was confirmed by the characteristic bands in the range of 3500–3000 cm−1 (νOH and νNH), 1610 cm−1 (amide band I) and 1570 cm−1 (amide band II), and C=O and C=C stretches in the range of 1700-1600 cm−1.”
Lines 190 – 193: “The average dimeter was assessed using the ImageJ software and expressed as mean ± standard deviation. Hence, a significant increase of the average fiber diameters was ob-served for PLA-HAP-Doxy sample compared to PLA-HAP sample.”
- Abstract can be improved, removal of Background, Method, etc., to improve readability and make it more attractive for readers to follow.
The section names (Background, Methods etc.) included in the abstract were removed (lines 27, 30, 35, 38, 39).
- Other minor things:
a) when referring to samples in Table, should be clear which is the tested and which is the control sample
Table 2 was modified, and the headings now contain information about the composition of each sample. The same information was added to lines 218, 219, 224 and 225).
b) minor English corrections are needed.
The article was reviewed from a language point of view.
c) Introduction: the emphasis of the aim of the present study can be improved - design of biomaterial was already optimized in a previous work, and now the focus in for the optimal sample, the in vitro antimicrobial effect.
For a more comprehensive aim statement of the study, the final paragraph in the introduction was changed to (lines 105-106):
“The aim of the present study is to assess the in vitro antimicrobial effect of a previously characterised and evaluated novel biomaterial based on PLA and nano-HAP nano-fibers, loaded with Doxycycline obtained by electrospinning against two periodontal pathogens: A. actinomycetemcomitans and P. gingivalis.”
We are looking forward to the assessment of the modified manuscript.
Sincerely,
The authors

Round 2
Reviewer 1 Report
In page 3- It would be a good idea to put materials details in the start of methodology chapter. this is completely missing here.
In page 3 line 109- No clear idea about the area density of nano fibreous mat is provided. It may have a different effect on the performance of DOXY dose. it should be included here
Is it possible to provide a schematic diagram of the development of this novel mat?
Page: mention the future scope of using Doxy based fibre mat in antimicrobial application
Author Response
Dear sir/madam,
We would like to thank you for the second revision of our submitted paper to the special issue in the journal Materials, “Advanced Smart Biomaterials and Techniques for Oral, Hard Tissue Engineering and Regeneration”.
After analysing your comments and suggestions, the following modifications have been made:
In page 3- It would be a good idea to put materials details in the start of methodology chapter. this is completely missing here.
The details of the material are specified in the Methodology chapter (Lines 111-118):
“Two different samples of nanofibers were obtained using the electrospinning tech-nique. The control sample was based on a 5% solution of PLA and nano-HAP. The test sample was based on the same PLA and nano-HAP solution, loaded with 7% Doxy by physical adsorption. The PLA-HAP and PLA-HAP-Doxy7 (Doxy-loaded HAP sample en-capsulated in PLA) nanofibers containing 5% PLA, a PLA/HAP ratio of 0.8 and 7 wt.% Doxy were prepared by electrospinning using a Fluidnatek LE-50 benchtop line (Bioinicia S.L., Valencia, Spain) by applying a voltage, flow-rate and tip-to-collector distance set at 27 kV, 1 mL/h and 15 cm as described in details in our previous paper [27].”
In page 3 line 109- No clear idea about the area density of nano fibreous mat is provided. It may have a different effect on the performance of DOXY dose. it should be included here
While we agree that the density of the nanofibers may influence its Doxy release, the parameter was not investigated when the material was characterised. The only information about the individual fibres was deduced through SEM, and it reveals the medium diameter of the fibres (lines 192-194):
“According to SEM, the average diameter of PLA-HAP nanofiber (310 ± 12 nm) increased significantly to 363 ± 24 nm upon the addition (encapsulation) of Doxy in case of PLA-HAP-Doxy7”
Is it possible to provide a schematic diagram of the development of this novel mat?
A schematic diagram was developed regarding the development of the material and its characterisation. The following sentence, image and image description were added (lines 123-127):
“The manufacturing and characterisation process of the nanofibers are represented schematically in Figure 1.
[Please see the attachment.]
Figure 1. Schematic representation of the technological manufacturing process of the nanofibers through electrospinning and their characterization.”
Page: mention the future scope of using Doxy based fibre mat in antimicrobial application
The following sentences were added (Lines 364 – 369):
“The developed material could be of use following the manual and ultrasonic debridement of periodontal pockets, in cases of aggressive bacterial populations, recurring infections or previous unsatisfactory therapeutic results. The material could be placed in the treated pockets, releasing the Doxy over a longer period, and helping with the stabilisation of the blood cloth, conditions needed for the reduction of the pocket depth and improvement of the clinical parameters.”
We are looking forward to hearing from you soon.
Sincerely,
The authors

Reviewer 2 Report
The authors answered all the reviewer questions and made modifications into the manuscript. It would have been better to introduce also an overview figure for sample characterization (even though most of this data was included in the authors previous work), however with the improved discussion with respect to the sample characterization authors now provide enough background to the readers, with respect to the samples morphology and composition.
Author Response
Dear sir/madam,
We would like to thank you for the second revision of our submitted paper to the special issue in the journal Materials, “Advanced Smart Biomaterials and Techniques for Oral, Hard Tissue Engineering and Regeneration”.
After analysing your comments and suggestions, the following modifications have been made:
The authors answered all the reviewer questions and made modifications into the manuscript. It would have been better to introduce also an overview figure for sample characterization (even though most of this data was included in the authors previous work), however with the improved discussion with respect to the sample characterization authors now provide enough background to the readers, with respect to the samples morphology and composition.
A schematic diagram was developed regarding the development of the matrial and ints characterisation. The following sentence, image and image description were added (lines 123-127):
“The manufacturing and characterisation process of the nanofibers are represented schematically in Figure 1.
[Please see the attachment]
Figure 1. Schematic representation of the technological manufacturing process of the nanofibers through electrospinning and their characterization.”
We are looking forward to hearing from you soon.
Sincerely,
The authors
